# Factors that determine women's autonomy to make decisions about sexual and reproductive health and rights in Nepal: A cross-sectional study

**Adweeti Nepal**[1]*, **Santa Kumar Dangol**[1], **Sujan Karki**[2], **Niraj Shrestha**[3]

**1** Health Programme, CARE Nepal, Lalitpur, Bagmati Province, Nepal, **2** Institute for Population and Social Research, Mahidol University, Nakhon Pathom, Thailand, **3** Abt Associates Inc., Kathmandu, Nepal

* anepal7@gmail.com

⊙ OPEN ACCESS

**Data Availability Statement:** This study is a secondary analysis of the data of NDHS 2016. The data sets are publicly available from The DHS

## Abstract

Women's autonomy on sexual and reproductive health issues is critical to women's health and well-being. Women have the right to decide on their fertility and sexuality, be free from coercion and violence, and achieve well-being. This study has identified women's autonomy regarding decision and exercise of their sexual reproductive health and rights and its association with determining factors in Nepal. Descriptive and analytical statistics such as bivariate and multivariate regression analysis were performed using data from Nepal Demographic and Health Survey 2016. The survey collected data from 12,862 women of reproductive age groups i.e. 15–49 years. However, for this study, we analyzed the data of only ever-married women and they were 9,875 in total. The analysis showed that women's autonomy in exercising their sexual reproductive health rights is highly associated with media exposure after controlling demographic variables. The frequency of exposure to media (*i. less than a week*: adjusted odds ratio (AOR):1.383; confidence interval (CI):1.145–1.670, p<0.001, *ii. at least once a week*: AOR:1.657; CI:1.359–2.021, p<0.001) is positively associated with women's autonomy. Furthermore, factors like women from Janajati (AOR:1.298; CI:1.071–1.576, p<0.01) and other Terai ethnic groups (AOR:1.471; CI:1.160–1.866, p<0.01), higher education attainment (AOR:1.482; CI:1.164–1.888, p<0.01), richest wealth quintile (AOR:1.527; CI:1.151–2.026, p<0.01), paid work (AOR:1.277; CI:1.045–1.561, p<0.05) and living in Lumbini Province (AOR:0.622; CI:0.486–0.797, p<0.001) and Sudur Paschim Province (AOR:0.723; CI:0.554–0.944, p<0.05) were found to be significantly associated with women's autonomy in sexual and reproductive health decision making. Similarly, women's autonomy is also increased with their increased age. In conclusion, women's exposure to media, improved socio-economic status and increased age influence their autonomy to make decisions about sexual and reproductive health rights in Nepal. Therefore, this study underscores the need to address socio-economic barriers and improve women's exposure to the media to enhance their autonomy further.

program's website https://www.dhsprogram.com/data/available-datasets.cfm.

**Funding:** The authors received no specific funding for this work.

**Competing interests:** The authors have declared that no competing interests exist.

## Introduction

Autonomy is a person's capacity to self-govern and act independently, responsibly, and with conviction [1]. In this case, the person's capacity refers to a woman's ability to exercise sexual and reproductive health and rights (SRHR). Various autonomy theories have been developed, understood, and applied in different ways in practice under the influence of laws, politics, philosophy, and religious precepts [2]. As per the bioethical principle, the capacity of a person's autonomy and right to direct their own life needs to be respected [3].

Women's autonomy on SRHR enables them to decide whether or not to participate in a sexual relationship with their husband or partner, their ability to decide on the use of contraception, and their rights to make independent decisions to seek and access sexual and reproductive health services [4–8]. In line with the commitments made in the 1994 International Conference on Population and Development (ICPD), which emphasized women's empowerment and reproductive rights issues, Nepal has taken SRHR as post-2015 agenda for translating rights from rhetoric to practice [9–11]. Moreover, Sustainable Development Goals (SDGs) five underscores the need to address structural barriers by altering the disproportionate power relations between women and men to ensure universal access to SRHR by 2030. It outlines that women have the right to live free of discrimination and violence and control and decide freely on matters related to their sexuality, including sexual and reproductive health (SRH) [12]. The Government of Nepal (GoN) has ratified the goals and adopted them as per the national context [13]. Following the ratification, GoN enacted the safe motherhood and reproductive health act in 2018, acknowledging the roles of reproductive health care in the lives of women and girls [14].

Despite policy provisions, the access to and utilization of sexual and reproductive health services is predominantly low among women of reproductive age, particularly those with low socio-economic status [15, 16]. Table 1 outlines that the coverage of sexual and reproductive health services:- antenatal care, institutional delivery, postnatal care, and family planning services; is comparatively lower for women from the lowest wealth quintile, having no or low education status and living hard-to-reach than women from highest wealth quintile, having secondary level education and residing in urban areas [16].

Study shows that age, education, husband's education, place of residence (rural/urban), and socio-economic factors are highly associated with women's empowerment on health-related

**Table 1. Overview of SRHR status in Nepal by background characteristics.**

| SRH status | Total | Wealth quintile | | Education | | Residence | |
|---|---|---|---|---|---|---|---|
| | | Highest | Lowest | Above secondary level | No education | Urban | Rural |
| The median age of marriage among women of 25–49 years (years) | 17.9 | 19.5 | 17.5 | 21.4 | 16.8 | 18.3 | 17.2 |
| Wanted fertility rate (rate) | 1.7 | 1.3 | 2.0 | 1.6 | 2.3 | 1.5 | 2.1 |
| Birth intervals (months) | 36.7 | 47.5 | 35.9 | 42.7 | 35.2 | 40.5 | 33.5 |
| Unmet need for family planning (%) | 23.7 | 20.5 | 27.0 | 25.0 | 17.9 | 22.7 | 25.3 |
| Teenage pregnancy (%) | 16.7 | 5.9 | 19.5 | 7.2 | 32.6 | 13.2 | 22.3 |
| Antenatal care coverage (%) | 83.6 | 95.5 | 73.8 | 94.5 | 73.3 | 87.0 | 79.5 |
| Institutional delivery coverage (%) | 57.4 | 89.6 | 33.9 | 85.4 | 36.4 | 68.6 | 44.2 |
| Delivery assisted by a skilled provider (%) | 58.0 | 88.7 | 33.9 | 84.9 | 37.6 | 67.7 | 46.8 |
| Postnatal care coverage (%) | 56.7 | 81.2 | 36.7 | 80.2 | 41.7 | 63.9 | 48.4 |
| Women experience physical violence (%) | 21.8 | 13.6 | 20.9 | 7.7 | 34.4 | 20.5 | 23.8 |
| Women experience sexual violence (%) | 6.9 | 6.6 | 7.7 | 3.8 | 9.2 | 7.1 | 6.6 |

Nepal Demographic and Health Survey (NDHS) 2016 [15].

issues [17]. Moreover, evidence suggests that women from developing countries like Nepal do not have the freedom to decide on the utilization of SRH services on their own because of harmful and discriminatory social norms and practices and a lack of financial resources [17–19]. Although there is some evidence in Nepal on women's autonomy in accessing maternal health services and family planning services, those studies have not analyzed the factors affecting women's autonomy in decision-making about SRHR as outlined by SDGs 5: indicator 5.6.1 [8, 20–22]. Therefore, this study aims to identify women's autonomy in decision-making for SRHR and its association with determining factors.

## Research methodology

The data set of Nepal Demographic and Health Survey (NDHS) 2016 has been used for the analysis. The data is obtained from https://www.dhsprogram.com/data/available-datasets.cfm with a registered data user. In the NDHS 2016, the data were collected from 12,862 women of reproductive age groups. From the perspective of our study, we analyzed the data of only ever-married women from the dataset, which was 9,875 women.

As stated in the NDHS 2016 published report, the sample was stratified and selected in two stages in rural areas and three stages in urban areas. In rural areas, wards were selected as primary sampling units (PSUs), and households were selected from the sample PSUs. Similarly, one enumeration area (EA) was selected from each PSU, and households were then selected from the sample EAs in urban settings. Each province was stratified into urban and rural areas, with 14 sampling strata from seven Provinces. The samples of wards were selected independently in each stratum.

NDHS 2016 used a stratified sampling design; it used a two-stage design in rural areas and a three-stage design in urban areas [23]. First, wards of the then village development committee and municipalities were considered PSUs. Due to the bigger size of wards of municipalities in terms of population density, from each PSU, one EA was selected. Then the household was selected from EA in urban areas and from PSU in rural areas.

In the first stage of survey, 383 wards were selected with probability proportional to ward size selected independently in each sampling stratum. The survey used a sampling frame of the 2011 national population census survey. The adjustment was made in the sampling frame due to population size changes from 2011 to 2015 throughout the stratum. After selecting the EA in urban areas and PSU in rural areas, a fixed number of households were selected with probability proportional to the size from the created household listing for data collection. No household replacement or changes in the pre-selected household was done to minimize bias in the implementation stages.

The sample was not self-weighting due to non-proportion sample allocation. Oversampling was done for fewer population Provinces. Weighting factors were calculated and added to the data file. The weighting factors were to be adjusted before the data analysis to represent the national, provincial, and regional levels.

### Definition and measurement of variables

**Dependent variable.** In this study, women's autonomy in decision-making for SRHR is taken as a dependent variable, which might be affected by changes in the value of independent variables. The dependent variable is defined as a proportion of women aged 15–49 years who make their own informed decision regarding (1) sexual relations, (2) contraceptive use, and (3) reproductive health care (Table 2). Hereafter, "women's autonomy" is used for "women's autonomy in decision-making for SRHR", throughout the paper.

**Table 2. Definition and measurement of the dependent variable.**

| Dependent variable | Definition and measurement |
|---|---|
| **Women's autonomy in decision-making for SRHR** | Women's autonomy is defined as the capacity of women aged 15–49 years old in decision-making and exercising their sexual and reproductive health rights making their own informed decisions regarding sexual relations, contraceptive use, and reproductive health care [15]. In NDHS 2016, responses to three specific questions were used to measure the level of autonomy in decision-making and exercise of their SRHR: <br> 1. Can you say no to your (husband/partner) if you do not want to have sexual intercourse? <br> 2. Would you say that using contraception is mainly your decision, your (husband's/partner's) decision, or did you both decide together? <br> 3. Who usually makes decisions about health care for yourself? <br> Women are considered to be autonomous if they say "yes" to question 1, "your decision" to question 2, and "you" to question 3 [15]. This is the composite indicator of decision-making about sexual intercourse with their husband or partner, decision for using contraceptive, and decision-making in receiving reproductive health care services [15, 21, 22]. |

**Independent variable.** The study selected different socio-economic characteristics as the dependent variable based on previous studies and consideration of NDHS 2016. Mass media is one of the primary sources of health information for women in Nepal [15]. Therefore, exposure to media has been considered as an independent variable for explaining the autonomy of women along with the other variables, such as age, religion, caste/ethnicity, education, place of residence in terms of the Province and urban/rural, wealth quintiles, occupation and size of the household as given in Table 3.

## Methods of data analysis

The analysis was done in three stages. Firstly, a univariate analysis was done to show the frequency distribution of dependent and independent variables. Secondly, bivariate analysis was carried out to assess the association between dependent and independent variables, where the study used the Chi-Square ($\chi^2$) test. Finally, a multivariate analysis was done to get an adjusted odds ratio (AOR) at 95% confidence interval (CI) to assess whether the socio-demographic factors are significant to women's autonomy in decision-making to exercise their SRHR in Nepal. Stata 14.2 statistical data analysis software is used for the analysis of the data.

## Ethical approval

This study is a secondary analysis of the data of NDHS 2016. The data sets are publicly available from the Demographic and Health Survey (DHS) program's website https://www.dhsprogram.com/data/available-datasets.cfm. The NDHS had obtained ethical approval from the Nepal Health Research Council and the ethical review board of ICF Macro International to conduct the survey. Prior verbal informed consent was taken from the participants for each interview. And for the participant under 18 years written informed consent was obtained from the parent/guardian for the interview.

## Results

Altogether 9,875 currently married women of the reproductive age group (15–49 years) were selected to determine the level of women's autonomy in this study. Among the participants, most of them were from 35–49 years groups (39%), followed by 25–29 years (20%) and 30–34 years (18%). Majority of the participants were from the Hindu (87%) community. More than

**Table 3. Definition and measurement of independent variables.**

| Independent variable | Definition and measurement |
|---|---|
| Exposure to media | It is a composite variable derived from the frequency of access to media; radio, television, and newspaper grouped into; no exposure, less than once a week, and at least once a week. |
| Age groups (in years) | Self-reported age of the women at the time of the survey, grouped into; <20 years, 20–24 years, 25–29 years, 30–34 years and 35–49 years for analysis. |
| Religion | Self-reported religious affiliation of the respondent; grouped into Hindu and other religions. Other religions cover; Muslim, Buddhist, Kirat, and Christian. |
| Caste/ethnicity | Self-reported caste/ethnic affiliation of respondents grouped into Dalit (so-called lower caste), Janajati (indigenous groups), and other Terai caste (other than Terai Dalit/Janajati/Brahmin Chhetri), Brahmin/Chhetri (so-called higher caste), Muslim, and others. The other includes; Marwadi, Bangali, Sikh, Jain, Panjabi, and other unidentified castes/ethnic groups. |
| Level of education | This is the attainment of the education by women which is grouped into; no education: who have never been to school, basic education: who completed the grade 10 and higher education: who studies more than 10$^{th}$ grade. |
| Husband's education | This is the attainment of the education by husband as reported by women who are grouped into; no education: who have never been to school, basic education: who completed the grade 10 and higher education: who studies more than 10$^{th}$ grade. |
| Place of residence by the Province | This is the place of residence of participants by Province; Province 1, Madhesh, Bagmati, Gandaki, Lumbini, Karnali, and Sudur Paschim. |
| Place of residence | This is a type of residence; urban and rural. |
| Wealth quintiles | It is a composite index of the cumulative living standard of a household, which is categorized into; poorest (bottom 20%), poorer (20%), middle (middle 20%), richer (20%), and richest (top 20%) based on wealth index. |
| Household size | Household size is defined based on the number of persons living in the same house and grouped into; a household with four or fewer members and five or more. |
| Occupation | Occupation is classified based on the self-report types of work and grouped into; no occupation (housewife and currently out of a job), agriculture, and others for the analysis of the data. Other occupations are paid work other than the agriculture sector. |
| Husband occupation | Husband's occupation is classified based on the report of women and grouped into; no occupation (currently out of a job), agriculture, and others for the analysis. Other occupations are mainly paid work. |

one-third of the participants were from Janajati (35%), followed by Brahmin and Chhetri (31%) and other Terai caste groups (16%). Two in five participants (40%) did not have any formal education, and almost the same percentage (41%) participants had only primary education. Only 19% of participants had a higher level of education. Among the study participants, more participants were from Madhesh Province (22%), followed by Bagmati (19%), Lumbini (18%), Province 1 (17%), Gandaki (10%), Sudur Paschim (9%) and Karnali (6%). More than 3 in 5 participants (61%) were from urban areas. Most of the participants (58%) had more than five or more family sizes. Almost half of the study participants (49%) were engaged in agriculture occupation (Table 4).

## The association of women's autonomy with explanatory variables

Women's autonomy in decision-making and exercise of their SRHR is highly associated with exposure to the media. Equally, autonomy is also strongly associated with women's age and their socio-economic status like caste/ethnicity, education level, residence, wealth quintiles, household size, occupation of women, and husbands' occupation. However, women's autonomy was not found to be associated with their religion and husband's educational level (Table 5).

**Table 4. Demographic variables of the study participants.**

| Variables | Frequency | Percentage |
|---|---|---|
| **Media exposure** | | |
| Not at all | 1798 | 18.2 |
| Less than once a week | 2172 | 22.0 |
| At least once a week | 5905 | 59.8 |
| **Education** | | |
| No education | 3,984 | 40.4 |
| Basic education | 4,030 | 40.8 |
| Higher education | 1,861 | 18.8 |
| **Husband's education** | | |
| No education | 1635 | 16.6 |
| Basic education | 5087 | 51.6 |
| Higher education | 3130 | 31.8 |
| **Age groups** | | |
| <20 years | 704 | 7.1 |
| 20–24 years | 1684 | 17.1 |
| 25–29 years | 1957 | 19.8 |
| 30–34 years | 1726 | 17.5 |
| 35–49 years | 3803 | 38.5 |
| **Caste/Ethnicity** | | |
| Dalit | 1265 | 12.8 |
| Muslim | 504 | 5.1 |
| Janajati | 3408 | 34.5 |
| Other Terai | 1593 | 16.2 |
| Brahmin/Chhetri | 3073 | 31.1 |
| Other | 32 | 0.3 |
| **Religion** | | |
| Hindu | 8552 | 86.6 |
| Other | 1322 | 13.4 |
| **Household size** | | |
| 4 or less | 4150 | 42.0 |
| 5 or more | 5725 | 58.0 |
| **Wealth quintile** | | |
| Poorest | 1687 | 17.1 |
| Poorer | 1946 | 19.7 |
| Middle | 2088 | 21.2 |
| Richer | 2107 | 21.3 |
| Richest | 2047 | 20.7 |
| **Occupation** | | |
| No occupation | 3142 | 31.8 |
| Agriculture | 4802 | 48.6 |
| Others | 1931 | 19.6 |
| **Husband's occupation** | | |
| No occupation | 364 | 3.7 |
| Agriculture | 2161 | 21.9 |
| Others | 7350 | 74.4 |
| **Place of residence by the Province** | | |
| Province 1 | 1655 | 16.8 |

(*Continued*)

**Table 4.** (Continued)

| Variables | Frequency | Percentage |
|---|---|---|
| Madhesh | 2168 | 22.0 |
| Bagmati | 1920 | 19.4 |
| Gandaki | 950 | 9.6 |
| Lumbini | 1749 | 17.7 |
| Karnali | 586 | 5.9 |
| Sudur Paschim | 846 | 8.6 |
| **Place of residence** | | |
| Urban | 6031 | 61.1 |
| Rural | 3844 | 38.9 |
| **Total** | **9875** | **100.0** |

## Determinants of women's autonomy in decision-making and exercise of their SRHR

The study identified that only one-fourth (25.1%) of the Nepalese women were autonomous in making their SRHR decisions. The multivariate logistic regression analysis showed that women's autonomy in decision-making and exercise of their SRHR is highly associated with media exposure after controlling demographic variables. The women who have media exposure at least once a week (AOR:1.657; CI:1.359–2.021) are more likely to be autonomous in deciding on their own and exercising their SRHR than those who have less or do not have exposure to media. Women who had higher education (AOR:1.482; CI:1.164–1.888) were likely to have autonomy compared to the women who having no formal education. Also women from richest quintile (AOR:1.527; CI:1.151–2.026) found be autonomous than women from other wealth quintiles. Similarly, it is found that increased age also plays a role in women's autonomy in SRHR decision making. Women of 15–19 years (AOR:0.155; CI:0.105–0.228) are less likely to have autonomy in decision-making compared to the women of 35–49 years. However, women from caste/ethnic group like Janajati (AOR:1.298; CI:1.071–1.576) and other Terai (AOR:1.471; CI:1.160–1.866) were found to be more autonomous in SRHR decision making in compare to so called higher caste i.e. Brahmin/Chhetri. Meanwhile the association is not significant for the women from Dalit caste (AOR:1.108; CI:0.898–1.358) and Muslim (AOR:0.949; CI:0.614–1.468) (refer Table 6).

## Discussion

Our analysis discovered that only one in four married women is autonomous to make decisions about SRHR in Nepal. Autonomous women were the women who were empowered to exercise their SRHR decision making over consensual sexual relations, contraceptive use and access to sexual and reproductive health services. Exposure to media, higher level of education attainment, paid work, increased age of women, women from Janajati and other Terai caste groups and richest quintile, and place of residence in terms of the Province (Lumbini and Sudur Paschim) are significantly associated with the women's autonomy in exercising their SRHR. However the basic education of women, husband's education, religion, size of household, and place of residence in terms of urban vs rural do not have any significant association with women's autonomy.

Similar to our findings, the studies conducted in lower and middle-income countries of Sub-Saharan Africa and India identified that media exposure is highly associated with a higher level of women's empowerment [24–26]. The evidence shows women's exposure to mass

**Table 5. Results from a bivariate analysis of women's autonomy in decision-making and exercise of their SRHR.**

| Variables | Total | Non- autonomous (%) | Autonomous (%) | $\chi^2$ (p-value) |
|---|---|---|---|---|
| **Exposure to media** | | | | |
| Not at all | 1798 | 82.5 | 17.5 | 97.4 (p <0.001) |
| Less than once a week | 2172 | 77.6 | 22.4 | |
| At least once a week | 5905 | 71.6 | 28.4 | |
| **Education of women** | | | | |
| No education | 3984 | 74.2 | 25.8 | 43.6 (p<0.001) |
| Basic education | 4031 | 77.9 | 22.2 | |
| Higher education | 1860 | 70.0 | 30.0 | |
| **Husband's education** | | | | |
| No education | 1635 | 74.2 | 25.8 | 12.3 (p<0.05) |
| Basic education | 5087 | 76.3 | 23.7 | |
| Higher education | 3130 | 73.0 | 27.0 | |
| **Age groups** | | | | |
| <20 years | 704 | 93.4 | 6.6 | 434.3 (p<0.001) |
| 20–24 years | 1684 | 87.7 | 12.3 | |
| 25–29 years | 1957 | 77.0 | 23.0 | |
| 33–34 years | 1725 | 70.8 | 29.2 | |
| 35–49 years | 3804 | 66.6 | 33.4 | |
| **Caste/Ethnicity** | | | | |
| Dalit | 1264 | 79.6 | 20.4 | 57.5 (p<0.001) |
| Muslim | 504 | 85.4 | 14.6 | |
| Janajati | 3408 | 72.3 | 27.7 | |
| Other Terai caste | 1593 | 74.1 | 25.9 | |
| Brahmin/Chhetri | 3073 | 74.6 | 25.4 | |
| Others | 32 | 72.4 | 27.6 | |
| **Religion** | | | | |
| Hindu | 8552 | 74.3 | 25.7 | 11.7 (p<0.05) |
| Others | 1322 | 78.7 | 21.3 | |
| **Household size** | | | | |
| 4 or less | 4150 | 72.9 | 27.1 | 15.6 (p<0.001) |
| 5 or more | 5725 | 76.4 | 23.6 | |
| **Wealth quintile** | | | | |
| Poorest | 1687 | 81.3 | 18.7 | 142.7 (p<0.001) |
| Poorer | 1946 | 77.2 | 22.8 | |
| Middle | 2088 | 76.2 | 23.8 | |
| Richer | 2107 | 75.5 | 24.5 | |
| Richest | 2047 | 65.5 | 34.6 | |
| **Occupation** | | | | |
| No occupation | 3142 | 78.2 | 21.8 | 99.4 (p<0.001) |
| Agriculture | 4802 | 76.2 | 23.8 | |
| Others | 1931 | 66.3 | 33.7 | |
| **Husband's occupation** | | | | |
| No occupation | 364 | 73.3 | 26.7 | 26.5 (p<0.001) |
| Agriculture | 2161 | 70.8 | 29.2 | |
| Others | 7350 | 76.2 | 23.8 | |
| **Place of residence by the Province** | | | | |

*(Continued)*

**Table 5.** (Continued)

| Variables | Total | Non- autonomous (%) | Autonomous (%) | $\chi^2$ (p-value) |
|---|---|---|---|---|
| Province 1 | 1655 | 70.9 | 29.1 | 99.8 (p<0.001) |
| Madhesh | 2169 | 75.8 | 24.3 | |
| Bagmati | 1920 | 69.0 | 31.0 | |
| Gandaki | 950 | 73.9 | 26.1 | |
| Lumbini | 1749 | 79.8 | 20.2 | |
| Karnali | 586 | 82.3 | 17.7 | |
| Sudur Paschim | 846 | 79.6 | 20.4 | |
| **Place of residence** | | | | |
| Urban | 6031 | 72.3 | 27.7 | 54.7 (p<0.001) |
| Rural | 3844 | 78.9 | 21.1 | |
| **Total** | **9875** | **74.9** | **25.1** | |

media improves access to information which empowers them to take part in the decision-making process by changing social norms, behaviours, and practices [26]. The media positively influence a person's will, assets, knowledge, and power relation within the family and society to make them more independent in their action [27]. Meanwhile, in Nepal, mass media is a crucial means of delivering the health-related message and creating awareness [15]. In addition, the GoN has invested significant resources in health information, education, and communication to improve sexual and reproductive health status. Therefore, the women exposed to media could have largely been exposed to the SRHR-related messages, which may have resulted in greater autonomy. This may not be the case in the other countries where rigid social norms exist. Hence, further study is needed to explore the relationship between exposure to SRHR-related message contents and women's autonomy to exercise their SRHR.

The study showed that women with a higher education level are more likely to be empowered in practicing their SRHR. It is a known fact that education empowers women to move from weak positions of power to exercise their control in different walks of life, including reducing gender inequalities [28]. Well-educated women are more likely to be informed about their health and well-being and they can also voice their freedom of expression rights which further empower them to use their rights [29, 30]. Moreover, the study found that women with higher education had a higher level of influence in exercising their rights. A similar association was found in rural India and various developing countries [7, 30–32]. But in this study, it is found that the educational background of the husband has no association with the empowerment of women. It is subject to be studied to explore the reason behind it in the context of Nepal. Also need to research the women's autonomy level in exercising their rights when both partners have a higher and same level of education.

The study found a positive association between women's increased age and empowerment in exercising SRHR. While the status of newly married women in the household could be a possible reason for it. In the South Asian context, including Nepal, a newly married daughter-in-law has less power in decision-making hence they follow the primary decision-makers. Moreover, the gender norms are much stricter for younger women including fertility pressure than the older women. Therefore, they get empowered as they get older with added and extended responsibilities and can decide independently [17, 33]. Moreover, other studies also identified that older women are empowered in decision-making in developing countries because of their changed status and roles with the increased age [19, 24].

**Table 6. Results from a multivariate logistic regression analysis of women's autonomy in decision-making and exercise of their SRHR.**

| Variables | Adjusted Odds Ratio (AOR) | p-value |
|---|---|---|
| **Media exposure** | | |
| No exposure to media | Ref | |
| Less than once a week | 1.383 (CI:1.145–1.670) | **0.001** |
| At least once a week | 1.657 (CI:1.359–2.021) | **0.001** |
| **Education** | | |
| No education | Ref | |
| Basic Education | 1.017 (CI: 0.887–1.165) | 0.811 |
| Higher Education | 1.482 (CI: 1.164–1.888) | **0.002** |
| **Husband's education** | | |
| No education | Ref | |
| Basic Education | 0.965 (CI: 0.814–1.141) | 0.674 |
| Higher Education | 0.873 (CI: 0.708–1.077) | 0.204 |
| **Age groups** | | |
| 35–49 years | Ref | |
| 15–19 years | 0.155 (CI: 0.105–0.228) | **0.001** |
| 20–24 years | 0.278 (CI: 0.219–0.351) | **0.001** |
| 25–29 years | 0.600 (CI: 0.452–0.690) | **0.001** |
| 30–34 years | 0.814 (CI: 0.8700–0.945) | **0.001** |
| **Ethnicity** | | |
| Brahmin/Chhetri | Ref | |
| Dalit | 1.108 (CI: 0.898–1.358) | 0.337 |
| Muslim | 0.949 (CI: 0.614–1.468) | 0.816 |
| Janajati | 1.298 (CI: 1.071–1.576) | **0.008** |
| Other Terai | 1.471 (CI: 1.160–1.866) | **0.002** |
| Other | 0.828 (CI: 0.345–1.988) | 0.673 |
| **Religion** | | |
| Hindu | Ref | |
| Other religion | 0.7980 (CI: 0.609–1.046) | 0.102 |
| **Household size** | | |
| 4 or less | Ref | |
| 5 or more | 0.968 (CI: 0.863–1.086) | 0.582 |
| **Wealth quintile** | | |
| Poor | Ref | |
| Poorer | 1.149 (CI: 0.936–1.409) | 0.183 |
| Middle | 1.246 (CI: 0.987–1.573) | 0.064 |
| Richer | 1.178 (CI: 0.924–1.500) | 0.186 |
| Richest | 1.527 (CI:1.151–2.026) | **0.003** |
| **Occupation** | | |
| Agriculture | Ref | |
| No occupation | 0.900 (CI: 0.762–1.064) | 0.219 |
| Others | 1.277 (CI: 1.045–1.561) | **0.017** |
| **Husband's occupation** | | |
| Agriculture | Ref | |
| No occupation | 0.716 (CI: 0.529–0.979) | **0.031** |
| Others | 0.690 (CI: 0.590–0.807) | **0.001** |
| **Province** | | |

(*Continued*)

**Table 6.** (Continued)

| Variables | Adjusted Odds Ratio (AOR) | p-value |
|---|---|---|
| Province 1 | Ref | |
| Madhesh | 0.896 (CI: 0.704–1.141) | 0.372 |
| Bagmati | 0.887 (CI: 0.676–1.163) | 0.382 |
| Gandaki | 0.819 (CI: 0.607–1.106) | 0.193 |
| Lumbini | 0.622(CI: 0.486–0.797) | **0.001** |
| Karnali | 0.746 (CI: 0.545–1.021) | 0.067 |
| Sudur Paschim | 0.723 (CI: 0.554–0.944) | **0.017** |
| **Place of residence** | | |
| Urban | Ref | |
| Rural | 0.860 (CI: 0.728–1.015) | 0.074 |
| **Total** | **9852** | |

Bold Significant at p-value <0.05

In Nepal, people are discriminated against by their socio-cultural background, including caste/ethnicity, which influences women's social position and their decision-making ability to access sexual and reproductive health services [34, 35]. Despite the fact, that Dalit women are more likely to be autonomous in SRHR decision-making than the women from Brahmin/Chhetri though the association is not significant. A study conducted in 2016 revealed that women of so-called untouchable caste groups are more likely to take part in economic activities in communities that support them to be economically empowered than the higher caste Hindu women. The assets support and enable them to take part in the independent decision-making process [36]. Moreover, the women from Janajati and other Terai caste groups are more likely to be empowered in practicing their SRHR.

This analysis has shown women's religion does not appear as a causal factor in the context of Nepal for women's empowerment in decision-making which is also subject to be explored in future studies. Meanwhile, Muslim women were found to be twice fewer autonomous than Janajati and other Terai ethnic women but the result is not significant. In the context of Nepal, Muslim is consider under the both ethnicity and religion. Also, more than 86% of respondents were from the Hindu religion, while other religions were in nominal numbers, which might have altered the association of faith with women's autonomy. Therefore, further exploration of the determinants of women's autonomy among Muslim and other religious women is recommended.

Women's economic empowerment is central to realizing women's rights and gender equality, hence, with the advancement in women's economic status, health services utilization can increase as they have more financial freedom to decide independently [37]. Studies conducted in developing countries found that women's economic level influences women's autonomy in decision-making and exercising their power for their benefit [24, 30]. Despite the fact, this study found a significant association between women's autonomy and economic status only among the women of highest wealth quintile. Several other factors explain the association between women's economic status and decision-making autonomy on SRHR [33].

Research shows that employment opportunity for women increases their autonomy in decision-making because of financial freedom [33]. In developing countries such as Senegal, Indonesia, and Ethiopia, women having paid work were autonomous in decision-making to exercise their health rights [38–41]. Paid employment appears as one of the factors of women's empowerment in the decision-making process and taking action independently for sexual and

reproductive health services in Nepal. Moreover, the women whose husbands paid jobs less were less likely to be empowered to do so. Similar to these findings, a national-level study conducted in Maynmar identified a positive association between women's employment status and empowerment in deciding their health. In the same study, it was also revealed that the women whose husbands have paid jobs were less likely to be empowered and get involved in the decision-making process in the household [42].

The country recently adopted the Federal system during the NDHS 2016 data collection period, where 77 districts are clustered under 7 provinces [15]. Each district has a unique socio-economic context in Nepal, so this could be a possible explanation for the variations in the result in each province in relation to women's autonomy and her residence province. The women of Lumbini and Sudur Paschim are significantly less likely to be autonomous than women of Province 1. Hence, further studies have to be conducted to show the association between the variables.

Evidence has shown that urban women are more likely to be autonomous in SRHR decision-making in compare to the rural women [33, 43–45]. However, in this analysis the association is not significant and this might be due to improved access to media, an improved level of literacy among rural women and improved availability of sexual and reproductive health services [46].

## Limitation

The findings of this study are based on the data of the NDHS 2016. The factors that influence women's autonomy were measured based on given indicators, i.e. women's decisions regarding sexual relations, contraceptive use, and reproductive health care. However, SRHR includes many other dimensions and indicators beyond the three selected indicators in this study. The conclusions of this study, therefore, should be comprehended and interpreted considering these limitations.

Meanwhile, this study has addressed the quality concerns by using nationally representative data for analysis which were collected following a standard methodological process. Also, the analysis of women's autonomy in SRHR and its associated factors is based on the indicator set by SDGs, which would be a reference for policymakers at different levels to design the SRHR program relevant to their context.

## Conclusion

In Nepal, only one fourth of married women of reproductive ages are autonomous in sexual and reproductive health decision-making. Better exposure to media, educational attainment, paid work, increased age, and economic status were significantly associated with a higher level of women's autonomy in SRHR decision-making. Moreover, women from Janajati and the other Terai caste group were also more likely to be autonomous. Women's autonomy in SRHR decision-making also depend on her residency province. Based on the finding, it is critical to enhance women's exposure to media and empower them educationally and financially for their increased level of autonomy. Context-specific SRHR policy and programs needed to be designed in Nepal and those programs should be focused on adolescents and young women and address the social norms and socio-economic barriers to accessing sexual and reproductive health services. It is also crucial to evaluate and document the different dimensions of women's autonomy in SRHR decision-making for informed policy decision.

## Acknowledgments

We acknowledge the DHS program for availing the NDHS 2016 data for analysis and public use.

## Author Contributions

**Conceptualization:** Adweeti Nepal, Santa Kumar Dangol, Niraj Shrestha.

**Data curation:** Adweeti Nepal, Santa Kumar Dangol.

**Formal analysis:** Sujan Karki.

**Software:** Sujan Karki.

**Writing – original draft:** Adweeti Nepal, Santa Kumar Dangol.

**Writing – review & editing:** Adweeti Nepal, Santa Kumar Dangol, Niraj Shrestha.

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
