## [Decision Letter · Decision Letter 0]

14 Jun 2022

PGPH-D-22-00866

Factors that Determine Women’s Autonomy in Decision Making for their Sexual and Reproductive Health and Rights in Nepal: A Cross-Sectional Study

Dear Adweeti,

Thank you for submitting your manuscript to PLOS Global Public Health. After careful consideration, we feel that it has merit but does not fully meet PLOS Global Public Health’s publication criteria as it currently stands. Therefore, we invite you to submit a revised version of the manuscript that addresses the points raised during the review process.

We look forward to receiving your revised manuscript.

Kind regards,

Collins Otieno Asweto, PhD

Academic Editor

Journal Requirements:

1. Please amend your detailed Financial Disclosure statement. This is published with the article, therefore should be completed in full sentences and contain the exact wording you wish to be published.

2. Please update the 'Competing Interests' statement with this "The authors have declared that no competing interests exist".

Reviewers' comments:

Reviewer's Responses to Questions

**Comments to the Author**

1. Does this manuscript meet PLOS Global Public Health’s publication criteria? Is the manuscript technically sound, and do the data support the conclusions? The manuscript must describe methodologically and ethically rigorous research with conclusions that are appropriately drawn based on the data presented.

Reviewer #1: Yes

Reviewer #2: Yes

2. Has the statistical analysis been performed appropriately and rigorously?

Reviewer #1: Yes

Reviewer #2: I don't know

3. Have the authors made all data underlying the findings in their manuscript fully available (please refer to the Data Availability Statement at the start of the manuscript PDF file)?

Reviewer #1: Yes

Reviewer #2: Yes

4. Is the manuscript presented in an intelligible fashion and written in standard English?

Reviewer #1: Yes

Reviewer #2: No

5. Review Comments to the Author

Reviewer #1: Overall the secondary analysis was conducted correctly with interesting findings. However some of findings can be a basis for the further studies. For instance on p.14 ratio of autonomous Muslims is twice less than ratio of autonomous Janajati. The line 159 and 160 states on absence of any association of autonomy with religion. Interesting finding is that higher education of women has more positive impact on autonomy than high education of their husbands. Could be interesting to see the cases where both are highly educated. Exposure to the media is associated with autonomy but it could be one of the possible factors among few which influences these women autonomy. Perhaps can be other factors along with exposure to media which leads to the greater autonomy. Impact form exposure to media depends on the context which media has and in Nepal mass media probably does talk about women rights. Unfortunately its not the case in all countries to the same extend and authors can be cautious with replication of these recommendation to other country contexts.

Reviewer #2: The study entitled “Factors Determining Women’s Autonomy in Decision Making for their Sexual and Reproductive Health and Rights” by Adweeti et al. discusses the factors that contribute to women’s autonomy in decision making on their sexual and reproductive health and rights (SRHR). The authors have written on a very important and interesting topic. However, some points should be considered in the revision of the manuscript:

1. There are some typographical and grammatic errors within the text that require correction to improve legibility.

2. The introduction outlines the progress and commitments made by Nepal in improving women’s decision making in their sexual and reproductive health and rights. The authors should consider adding additional literature on how their study relates to previously published research on women’s decision making on their SRHR.

3. The authors should consider checking the measurement of the dependent variable and the cited reference [15] which does not have the questions used to measure autonomy.

6. PLOS authors have the option to publish the peer review history of their article (what does this mean?). If published, this will include your full peer review and any attached files.

**Do you want your identity to be public for this peer review?** For information about this choice, including consent withdrawal, please see our Privacy Policy.

Reviewer #1: **Yes: **Dr. K. Goshliyev

Reviewer #2: No

---

## [Decision Letter · Decision Letter 1]

5 Jul 2022

Factors that Determine Women’s Autonomy to Make Decisions About Sexual and Reproductive Health and Rights in Nepal: A Cross-Sectional Study

PGPH-D-22-00866R1

Dear Adweeti

We are pleased to inform you that your manuscript 'Factors that Determine Women’s Autonomy to Make Decisions About Sexual and Reproductive Health and Rights in Nepal: A Cross-Sectional Study' has been provisionally accepted for publication in PLOS Global Public Health.

Best regards,

Collins Otieno Asweto, PhD

Academic Editor

Reviewer Comments (if any, and for reference):

Reviewer's Responses to Questions

**Comments to the Author**

1. If the authors have adequately addressed your comments raised in a previous round of review and you feel that this manuscript is now acceptable for publication, you may indicate that here to bypass the “Comments to the Author” section, enter your conflict of interest statement in the “Confidential to Editor” section, and submit your "Accept" recommendation.

Reviewer #2: All comments have been addressed

2. Does this manuscript meet PLOS Global Public Health’s publication criteria? Is the manuscript technically sound, and do the data support the conclusions? The manuscript must describe methodologically and ethically rigorous research with conclusions that are appropriately drawn based on the data presented.

Reviewer #2: Yes

3. Has the statistical analysis been performed appropriately and rigorously?

Reviewer #2: I don't know

4. Have the authors made all data underlying the findings in their manuscript fully available (please refer to the Data Availability Statement at the start of the manuscript PDF file)?

Reviewer #2: Yes

5. Is the manuscript presented in an intelligible fashion and written in standard English?

Reviewer #2: Yes

6. Review Comments to the Author

Reviewer #2: (No Response)

7. PLOS authors have the option to publish the peer review history of their article (what does this mean?). If published, this will include your full peer review and any attached files.

**Do you want your identity to be public for this peer review?** For information about this choice, including consent withdrawal, please see our Privacy Policy.

Reviewer #2: No
